# A Flexible Generative Framework for Graph-based Semi-supervised Learning

**Jiaqi Ma**[*†]
jiaqima@umich.edu

**Weijing Tang**[*‡]
weijtang@umich.edu

**Ji Zhu**[‡]
jizhu@umich.edu

**Qiaozhu Mei**[†§]
qmei@umich.edu

## Abstract

We consider a family of problems that are concerned about making predictions for the majority of unlabeled, graph-structured data samples based on a small proportion of labeled samples. Relational information among the data samples, often encoded in the graph/network structure, is shown to be helpful for these semi-supervised learning tasks. However, conventional graph-based regularization methods and recent graph neural networks do not fully leverage the interrelations between the features, the graph, and the labels. In this work, we propose a flexible generative framework for graph-based semi-supervised learning, which approaches the joint distribution of the node features, labels, and the graph structure. Borrowing insights from random graph models in network science literature, this joint distribution can be instantiated using various distribution families. For the inference of missing labels, we exploit recent advances of scalable variational inference techniques to approximate the Bayesian posterior. We conduct thorough experiments on benchmark datasets for graph-based semi-supervised learning. Results show that the proposed methods outperform the state-of-the-art models in most settings.

## 1  Introduction

Traditional machine learning methods typically treat data samples as independent and approximate a mapping function from the features to the outcome of each individual sample. However, many real-world data, such as social media or scientific articles, often come with richer relational information among the individual samples. We consider a family of such scenarios where the relational information is stored in a graph structure with the data samples as nodes, and the learning task is to predict the outcomes of unlabeled nodes based on the node features, the graph structure, as well as the labels of a subset of nodes. In these scenarios, breaking the independence assumption and utilizing such relational information in the prediction models have been shown to be helpful [26, 25, 1, 10, 5, 14, 12]. However, there lacks a principled way to best synergize and utilize the relational information stored in the graph together with the information stored in individual nodes. In this paper, we consider the problem of graph-based semi-supervised learning and try to approach this problem by presenting a flexible generative framework that explicitly models the joint relationship among the three key types of information in this context: features, outcomes (or labels), and the graph.

There are two major classes of existing methods for graph-based semi-supervised learning. The first class includes the graph-based regularization methods [26, 25, 1, 14, 12], where explicit regularizations are posed to smooth the predictions or feature representations over local neighborhoods. This class of methods share an assumption that some kind of smoothness (e.g., the outcomes of adjacent nodes are likely to be the same) should present in the local and global graph structure. The

---

[*]The two authors contribute equally to this paper.

[†]School of Information, University of Michigan

[‡]Department of Statistics, University of Michigan

[§]Department of EECS, University of Michigan

second class consists of graph neural networks [10, 5, 21], where the node features within a local neighborhood are aggregated into a hidden representation for the ego node and predictions are made on top of the hidden representations. These existing methods either do not treat the graph as a random variable (but rather as a fixed observation) or do not jointly model the data features, graph, and outcomes.

While having not been well-explored in graph-based semi-supervised learning, we believe that modeling the joint distribution of the data, graph, and labels with generative models has several unique advantages over the above methods.

First, generative models can learn succinct underlying structures of the graph data. Rich literature in network science [16] has shown that underlying structures often exist in real-world graph data. And there have been many probabilistic generative models [7, 6] that can learn the underlying structures well from observed graph data. Most of the existing graph-based semi-supervised learning methods described above view the graph as a fixed observation and treat it as ground truth. In reality, however, an observed graph is often noisy. We expect that through treating features, outcomes, and graph as random variables, a generative model can capture more general patterns among these entities and learn low-dimensional representations of the data that can take account for the noise in the graph.

Second, modeling the joint distribution can extract more general relationship among features, outcomes, and the graph. We argue that both classes of the existing graph-based semi-supervised learning methods only utilize restricted relationships among them. The graph-based regularization methods usually make strong assumptions about smoothness over adjacent nodes. Such assumptions often restrict the model capacity, making the models fail to fully utilize the relational information. The graph neural networks, although more flexible in aggregating node features through the graph structure, usually implicitly assume conditional independence over the outcomes given node features and the graph. This might be sub-optimal in utilizing the relational information. Directly modeling the joint distribution with flexible models allows us to better utilize the relational information.

Moreover, generative models can better handle the missing-data situation. In real-world applications, we are often faced with imperfect data, where either node features or edges in the graph are missing. Generative models excel in such situations.

A few previous studies [24] trying to apply generative models to graph-based semi-supervised learning have been restricted to relatively simple model families due to the difficulty in efficient training of generative models. Thanks to recent advances of scalable variational inference techniques [9, 8], we are able to propose a flexible generative framework for graph-based semi-supervised learning. In this work, we use neural networks, latent space models [6], and stochastic block models [7] to form the generative models. And we use graph neural networks as the approximate posterior models in the scalable variational inference. We refer such instantiations of the proposed framework as $\text{G}^3\text{NN}$ (Generative Graph models with Graph Neural Networks as approximate posterior). We evaluate the proposed framework with four variants of $\text{G}^3\text{NN}$ on three semi-supervised classification benchmark datasets. Experiments show that our models achieve better performances than the state-of-the-art models under most settings.

## 2 Related Work

This paper mainly focuses on the problem of graph-based semi-supervised learning, where the data samples are connected by a graph and the outcome labels are only available for part of the samples. The goal is to infer the unobserved labels based on both the labeled and unlabeled data as well as the graph structure.

### 2.1 Graph-based Regularization for Semi-supervised Learning

One of the most popular types of graph-based semi-supervised learning methods is the graph-based regularization methods. The general assumption of such methods is that the data samples are located in a low-dimensional manifold where each local neighborhood is a high-dimensional Euclidean space, and the graph stores the similarity or proximity of these data samples. Various graph regularizations are posed to smooth the outcome predictions of the model or the feature representations of the data samples over the local neighborhood in the graph. Suppose there are $n$ data samples in total and $m$ of them are labeled, the graph-based regularization methods generally conduct semi-supervised learning

by optimizing the following objective function:

$$\sum_{i=1}^{m} \mathcal{L}_i + \eta \sum_{i,j=1}^{n} w_{i,j} \mathcal{R}(\boldsymbol{f}_i, \boldsymbol{f}_j),$$

where $\mathcal{L}_i$ is the supervised loss function of sample $i$; $\mathcal{R}(\cdot, \cdot)$ is a regularization function and $w_{i,j}$ is a graph-based coefficient; $\boldsymbol{f}_i, \boldsymbol{f}_j$ could be the outcome predictions [26, 25, 1] or the feature representations [14, 22, 12] of nodes $i$ and $j$; $\eta$ is a hyper-parameter trading-off the supervised loss and the graph-based regularization. Different methods can have different variants of the regularization term. Most commonly, it is set as a graph Laplacian regularizer [26, 25, 1, 14, 22]. Such type of models heavily rely on the smoothness assumption over the graph, which restricts the modeling capacity [10].

## 2.2 Graph Neural Networks for Semi-supervised Learning

Another class of methods that have gained great attention recently are the graph neural networks [10, 5, 21]. A graph neural network aggregates the node features within a local neighborhood into a hidden representation for the central node. Such aggregation operations can also be stacked on top of the hidden representations to form deeper neural networks. Generally, a single aggregation operation for node $i$ at depth $l$ can be represented as follows,

$$h_i^l = \sigma(\sum_{j \in \mathcal{N}_i} \alpha_{i,j} \boldsymbol{W} h_j^{l-1}),$$

where $h_i^l$ is the hidden representation of $i$ at $l^{\text{th}}$ layer; $\mathcal{N}_i$ is the neighbor set of $i$; $\boldsymbol{W}$ is a learnable linear transformation matrix; $\sigma$ is an element-wise nonlinear activation function; and different models have different definitions of $\alpha_{i,j}$. For Graph Convolutional Networks [10], $\alpha_{i,j} = 1/d_i$ or $\alpha_{i,j} = 1/\sqrt{d_i d_j}$, where $d_i$ is the number of neighbors of $i$. For Graph Attention Networks [21], $\alpha_{i,j}$ is defined as an attention function between $i$ and $j$. Finally, the predictions of each node are made on top of hidden representations in the last layer. Such methods usually model the mapping from the features and the graph to the outcome of an individual node, which assume the outcomes are conditionally independent given the features and the graph. This assumption prevents the model from utilizing the joint relationship among the outcomes over the graph. Our framework models the joint distribution of the features, outcomes, and the graph, and it is not restricted to this assumption. A concurrent work [18] also tries to mitigate this assumption. They take a statistical relational learning point of view and model the outcome dependency with a Markov network conditioned on the graph, while we take a generative model point of view and instantiate the joint distribution with random graph models.

## 2.3 Generative Methods for Graph-based Semi-supervised Learning

Most methods from the above two classes treat the graph as fixed observation and only a few methods [17, 24, 13] treat the graph as a random variable and model it with generative models. Among them, Ng et al. [17] focused more on an active learning setting on graphs; Zhang et al. [24] modeled the graph along with a stochastic block model and did not consider the interaction between the graph and the features or the labels in the generative model of the graph; Liu [13] shares the most similar generative model with our framework, but considers a supervised learning setting where the labels are fully observed. To our best knowledge, our work is the first to propose a generative framework for graph-based semi-supervised learning that models the joint distribution of features, outcomes, and the graph with flexible nonlinear models.

Finally, as a side note, there is a recently active area of deep generative models for graphs [11, 3]. But these models focus more on generating realistic graph topology structures and are less related to the graph-based semi-supervised learning, which is the problem of interest in this work.

# 3 Approach

## 3.1 Problem Setup

We start by formally introducing the problem of graph-based semi-supervised learning. Given a set of data samples $\mathcal{D} = \{(\boldsymbol{x}_i, \boldsymbol{y}_i)\}_{i=1}^{n}$, $\boldsymbol{x}_i \in \mathbb{R}^d$ and $\boldsymbol{y}_i \in \mathbb{R}^l$ are the feature and outcome vectors

of sample $i$ respectively. We further denote $\boldsymbol{X} \in \mathbb{R}^{n \times d}$ and $\boldsymbol{Y} \in \mathbb{R}^{n \times l}$ as the matrices formed by feature and outcome vectors. The dataset also comes with a graph $G = (\mathcal{V}, \mathcal{E})$ with the data samples as nodes, where $\mathcal{V} = \{1, 2, \cdots, n\}$ is the set of nodes and $\mathcal{E} \subseteq \mathcal{V} \times \mathcal{V}$ is the set of edges. In the semi-supervised learning setting, only $0 < m < n$ samples have observed their outcome labels and the outcome labels of other samples are missing. Without loss of generality, we assume the outcomes of the samples $1, 2, \cdots, m$ are observed and that of $m + 1, \cdots, n$ are missing. Therefore we can partition the outcome matrix as

$$\boldsymbol{Y} = \left[ \begin{array}{c} \boldsymbol{Y}_{\mathrm{obs}} \\ \boldsymbol{Y}_{\mathrm{miss}} \end{array} \right].$$

The goal of graph-based semi-supervised learning is to infer $\boldsymbol{Y}_{\mathrm{miss}}$ based on $(\boldsymbol{X}, \boldsymbol{Y}_{\mathrm{obs}}, G)$. For discriminative methods, we are learning the conditional distribution of $p(\mathbf{Y}|\mathbf{X}, \mathbf{G})$. This is usually done by learning a prediction model $\boldsymbol{y} = f(\boldsymbol{x}; \boldsymbol{X}, G)$ using empirical risk minimization, optionally with regularizations:

$$\hat{f} = \arg \min_{f} \frac{1}{m} \sum_{i=1}^{m} \mathcal{L}(\boldsymbol{y}_i, f(\boldsymbol{x}_i; \boldsymbol{X}, G)) + \lambda \mathcal{R}(f; G),$$

where $\mathcal{L}(\cdot, \cdot)$ is a loss function, $\mathcal{R}(\cdot; G)$ is a graph-based regularization term, and $\lambda$ is a hyper-parameter controlling the strength of the regularization. Then $\hat{f}$ is used to predict $\boldsymbol{Y}_{\mathrm{miss}}$.

There are two specific learning settings, namely transductive learning and inductive learning, which are common in graph-based semi-supervised learning. Transductive learning assumes that $\boldsymbol{X}$ and $G$ are fully observed during both the learning stage and the inference stage while inductive learning assumes $\boldsymbol{X}_{m+1:n}$ and the nodes of $m + 1, \cdots, n$ in $G$ are missing during the learning stage but available during the inference stage. In the following of this paper, we will mainly focus on the transductive learning setting but our method can also be extended to the inductive learning setting.

## 3.2 A Flexible Generative Framework for Graph-based Semi-supervised Learning

In discriminative methods, the graph $G$ is usually viewed as a fixed observation. In reality, however, there is usually considerable noise in the graph. Moreover, we want to take advantage of the underlying structure among $\boldsymbol{X}, \boldsymbol{Y}$, and $G$ to improve the prediction performance under this semi-supervised learning setting. In this work, we propose a flexible generative framework that can model a wide range of the forms of the joint distribution $p(\mathbf{X}, \mathbf{Y}, \mathbf{G})$, where $\mathbf{X}, \mathbf{Y}$, and $\mathbf{G}$ are the random variables corresponding to $\boldsymbol{X}, \boldsymbol{Y}, G$. We also denote $\mathbf{Y}_{\mathrm{obs}}$ and $\mathbf{Y}_{\mathrm{miss}}$ as the random variable counterparts of $\boldsymbol{Y}_{\mathrm{obs}}$ and $\boldsymbol{Y}_{\mathrm{miss}}$.

**Generation process.** Inspired by the random graph models from the area of network science [16], we assume the graph is generated based on the node features and outcomes. The generation process can be illustrated by the following factorization of the joint distribution:

$$p(\mathbf{X}, \mathbf{Y}, \mathbf{G}) = p(\mathbf{G}|\mathbf{X}, \mathbf{Y})p(\mathbf{Y}|\mathbf{X})p(\mathbf{X}),$$

where the conditional probabilities $p(\mathbf{G}|\mathbf{X}, \mathbf{Y})$ and $p(\mathbf{Y}|\mathbf{X})$ will be modeled by some flexible parametric families distributions $p_{\boldsymbol{\theta}}(\mathbf{G}|\mathbf{X}, \mathbf{Y})$ and $p_{\boldsymbol{\theta}}(\mathbf{Y}|\mathbf{X})$ with parameters $\boldsymbol{\theta}$. By "flexible" we mean that the only restriction on the PMFs of these conditional probabilities is that they need to be differentiable almost everywhere w.r.t. $\boldsymbol{\theta}$; and we do not assume the integral of the marginal distribution $p_{\boldsymbol{\theta}}(\mathbf{G}|\mathbf{X}) = \int p_{\boldsymbol{\theta}}(\mathbf{Y}|\mathbf{X})p_{\boldsymbol{\theta}}(\mathbf{G}|\mathbf{Y}, \mathbf{X})d\mathbf{Y}$ is tractable. For simplicity, we do not specify the distribution $p(\mathbf{X})$ and everything will be conditioned on $\mathbf{X}$ later in this paper.

**Model inference.** To infer the missing outcomes $\mathbf{Y}_{\mathrm{miss}}$, we would need the posterior distribution $p_{\boldsymbol{\theta}}(\mathbf{Y}_{\mathrm{miss}}|\mathbf{X}, \mathbf{Y}_{\mathrm{obs}}, \mathbf{G})$, which is usually intractable under many flexible generative models. Following the recent advances in scalable variational inference [9, 8], we introduce a recognition model $q_{\boldsymbol{\phi}}(\mathbf{Y}_{\mathrm{miss}}|\mathbf{X}, \mathbf{Y}_{\mathrm{obs}}, \mathbf{G})$ parameterized by $\boldsymbol{\phi}$ to approximate the true posterior $p_{\boldsymbol{\theta}}(\mathbf{Y}_{\mathrm{miss}}|\mathbf{X}, \mathbf{Y}_{\mathrm{obs}}, \mathbf{G})$.

**Model learning.** We train the model parameters $\boldsymbol{\theta}$ and $\boldsymbol{\phi}$ by optimizing the Evidence Lower BOund (ELBO) of the observed data $(\boldsymbol{Y}_{\mathrm{obs}}, G)$ conditioned on $\boldsymbol{X}$. The negative ELBO loss $\mathcal{L}_{\mathrm{ELBO}}$ is defined as follows,

$$\log p(\boldsymbol{Y}_{\mathrm{obs}}, G|\boldsymbol{X}) \geq \mathbb{E}_{q_{\boldsymbol{\phi}}(\mathbf{Y}_{\mathrm{miss}}|\boldsymbol{X}, \boldsymbol{Y}_{\mathrm{obs}}, G)}(\log p_{\boldsymbol{\theta}}(\mathbf{Y}_{\mathrm{miss}}, \boldsymbol{Y}_{\mathrm{obs}}, G|\boldsymbol{X}) - \log q_{\boldsymbol{\phi}}(\mathbf{Y}_{\mathrm{miss}}|\boldsymbol{X}, \boldsymbol{Y}_{\mathrm{obs}}, G))$$
$$\triangleq -\mathcal{L}_{\mathrm{ELBO}}(\boldsymbol{\theta}, \boldsymbol{\phi}; \boldsymbol{X}, \boldsymbol{Y}_{\mathrm{obs}}, G).$$

And the optimal model parameters are obtained by minimizing the above loss:

$$\hat{\boldsymbol{\theta}}, \hat{\boldsymbol{\phi}} = \underset{\boldsymbol{\theta}, \boldsymbol{\phi}}{\arg\min}\, \mathcal{L}_{\mathrm{ELBO}}(\boldsymbol{\theta}, \boldsymbol{\phi}; \boldsymbol{X}, \boldsymbol{Y}_{\mathrm{obs}}, G).$$

## 3.3 G$^3$NN Instantiations

For practical use, it remains to specify the parametric forms of the generative model $p_{\boldsymbol{\theta}}(\mathbf{G}|\mathbf{X}, \mathbf{Y})$ and $p_{\boldsymbol{\theta}}(\mathbf{Y}|\mathbf{X})$, and the approximate posterior model $q_{\boldsymbol{\phi}}(\mathbf{Y}_{\mathrm{miss}}|\mathbf{X}, \mathbf{Y}_{\mathrm{obs}}, \mathbf{G})$. In this section, we instantiate the Generative Graph model with two types of random graph models, and adopt two types of Graph Neural Networks as the approximate posterior model, which leads to four variants of G$^3$NN. As proof of the effectiveness of our general framework, we intend to instantiate its components with simple models and leave room for optimizing its performance with more complex instantiations. The generative framework proposed above does not restrict the type of outcomes. As proof of concept, we focus on multi-class classification outcomes in the following of the paper and denote the number of classes as $K$.

### 3.3.1 Instantiations of the Generative Model

For $p_{\boldsymbol{\theta}}(\mathbf{Y}|\mathbf{X})$ in the generative model, we simply instantiate it with a multi-layer perceptron. For $p_{\boldsymbol{\theta}}(\mathbf{G}|\mathbf{X}, \mathbf{Y})$ in the generative model, we have come up with two instantiations inspired by the generative network models from network science literature. There are two major classes of generative models for complex networks: the latent space models (LSM) [6] and the stochastic block models (SBM) [7]. We instantiate a simple model from each class as our generative models.

A general assumption used by both classes of generative models for networks is that the edges are conditionally independent. Let $\mathrm{e}_{i,j} \in \mathcal{V} \times \mathcal{V}$ be the binary edge random variable between node $i$ and $j$. $\mathrm{e}_{i,j} = 1$ indicates the edge between $i$ and $j$ exists and $0$ otherwise. Based on the conditional independence assumption of edges, the conditional probability of the graph $\mathbf{G}$ can be factorized as

$$p_{\boldsymbol{\theta}}(\mathbf{G}|\mathbf{X}, \mathbf{Y}) = \prod_{i,j} p_{\boldsymbol{\theta}}(\mathrm{e}_{i,j}|\mathbf{X}, \mathbf{Y}).$$

Next we will specify the instantiations of $p_{\boldsymbol{\theta}}(\mathrm{e}_{i,j}|\mathbf{X}, \mathbf{Y})$.

**Instantiation with an LSM.** The latent space model assumes that the nodes lie in a latent space and the probability of $\mathrm{e}_{i,j}$ only depends the representation of nodes $i$ and $j$. i.e., $p_{\boldsymbol{\theta}}(\mathrm{e}_{i,j}|\mathbf{X}, \mathbf{Y}) = p_{\boldsymbol{\theta}}(\mathrm{e}_{i,j}|\mathbf{x}_i, \mathbf{y}_i, \mathbf{x}_j, \mathbf{y}_j)$. We assume it follows a logistic regression model:

$$p_{\boldsymbol{\theta}}(\mathrm{e}_{i,j} = 1|\mathbf{x}_i, \mathbf{y}_i, \mathbf{x}_j, \mathbf{y}_j) = \sigma([(\boldsymbol{U}\mathbf{x}_i)^T, \mathbf{y}_i^T, (\boldsymbol{U}\mathbf{x}_j)^T, \mathbf{y}_j^T]\boldsymbol{w}),$$

where $\sigma(\cdot)$ is the sigmoid function; $\boldsymbol{w}$ are the learnable parameters of the logistic regression model; $\boldsymbol{U}$ is a linear transformation matrix with learnable parameters (e.g., word embedding when $\mathbf{x}$ is a bag-of-words feature); the class labels $\mathbf{y}_i, \mathbf{y}_j$ are represented as one-hot vectors, and we concatenate the transformed features and the class labels of a pair of nodes as input of the logistic regression model. All the learnable parameters are included in $\boldsymbol{\theta}$.

**Instantiation with an SBM.** The stochastic block model assumes there are $C$ types of nodes and each node $i$ has a (latent) type variable $\mathrm{z}_i \in \{1, 2, \cdots, C\}$ and the probability of edge $\mathrm{e}_{i,j}$ only depends on node types $\mathrm{z}_i$ and $\mathrm{z}_j$. In a general SBM, we have $p_{\boldsymbol{\theta}}(\mathrm{e}_{i,j} = 1|\mathrm{z}_i, \mathrm{z}_j) = p_{\mathrm{z}_i, \mathrm{z}_j}$, and the $p_{u,v}$ for all $u, v \in \{1, 2, \cdots, C\}$ form a probability matrix $\boldsymbol{P}$ and are free parameters to be fitted.

In our model, we assume the node types are the class labels, i.e., $C = K$ and $\mathrm{z}_i$ being the corresponding class of $\mathbf{y}_i$, and $p_{\boldsymbol{\theta}}(\mathrm{e}_{i,j}|\mathbf{X}, \mathbf{Y}) = p_{\boldsymbol{\theta}}(\mathrm{e}_{i,j}|\mathbf{y}_i, \mathbf{y}_j)$. Note that in our notation $\mathbf{y}_i$ is a one-hot vector. We specify $p_{\boldsymbol{\theta}}(\mathrm{e}_{i,j}|\mathbf{y}_i, \mathbf{y}_j)$ with the simplest SBM, which is also called the planted partition model. That is,

$$\mathrm{e}_{i,j}|\mathbf{y}_i, \mathbf{y}_j \sim \left\{ \begin{array}{ll} \mathrm{Bernoulli}(\mathrm{p}_0) & \text{if } \mathbf{y}_i = \mathbf{y}_j \\ \mathrm{Bernoulli}(\mathrm{p}_1) & \text{if } \mathbf{y}_i \neq \mathbf{y}_j \end{array} \right. .$$

This means the probability matrix $\boldsymbol{P}$ has all the diagonal elements equal to a constant $p_0$ and all the off-diagonal elements equal to another constant $p_1$.

### 3.3.2 Instantiations of the Approximate Posterior Model

For the approximate posterior model $q_\phi(\mathbf{Y}_{\mathrm{miss}}|\mathbf{X}, \mathbf{Y}_{\mathrm{obs}}, \mathbf{G})$, in principle we need a strong function approximator that takes $(\mathbf{X}, \mathbf{Y}_{\mathrm{obs}}, \mathbf{G})$ as the input and outputs the probability of $\mathbf{Y}_{\mathrm{miss}}$. Here we consider two recently invented graph neural networks: the Graph Convolutional Network (GCN) [10] and the Graph Attention Network (GAT) [21]. Note that by doing this we are making a further approximation from $q_\phi(\mathbf{Y}_{\mathrm{miss}}|\mathbf{X}, \mathbf{Y}_{\mathrm{obs}}, \mathbf{G})$ to $q_\phi(\mathbf{Y}_{\mathrm{miss}}|\mathbf{X}, \mathbf{G})$, as the graph neural networks by design only take $(\mathbf{X}, \mathbf{G})$ as the input. This approximation is known as the mean-field method, which is commonly used in variational inference [2].

### 3.4 Training

Finally, we end this section by introducing two practical details of model training.

**Supervised loss.** As our main task is to conduct classification with the approximate posterior model, similar with Kingma et al. [8], we add an additional supervised loss to better train the approximate posterior model:
$$\mathcal{L}_s(\phi; \mathbf{X}, \mathbf{Y}_{\mathrm{obs}}, G) = -\log q_\phi(\mathbf{Y}_{\mathrm{obs}}|\mathbf{X}, G).$$
The total loss is controlled by a weight hyper-parameter $\eta$,
$$\mathcal{L}(\boldsymbol{\theta}, \phi) = \mathcal{L}_{\mathrm{ELBO}}(\boldsymbol{\theta}, \phi; \mathbf{X}, \mathbf{Y}_{\mathrm{obs}}, G) + \eta \cdot \mathcal{L}_s(\phi; \mathbf{X}, \mathbf{Y}_{\mathrm{obs}}, G).$$

We could rewrite the total loss in an alternative way as follows,
$$\begin{aligned}\mathcal{L}(\boldsymbol{\theta}, \phi) =& \mathbb{E}_{q_\phi(\mathbf{Y}_{\mathrm{miss}}|\mathbf{X}, G)} - \log p_{\boldsymbol{\theta}}(G|\mathbf{X}, \mathbf{Y}_{\mathrm{obs}}, \mathbf{Y}_{\mathrm{miss}}) + \mathcal{D}_{\mathrm{KL}}(q_\phi(\mathbf{Y}_{\mathrm{miss}}|\mathbf{X}, G)\|p_{\boldsymbol{\theta}}(\mathbf{Y}_{\mathrm{miss}}|\mathbf{X})) \\ &- \log p_{\boldsymbol{\theta}}(\mathbf{Y}_{\mathrm{obs}}|\mathbf{X}) - \eta \cdot \log q_\phi(\mathbf{Y}_{\mathrm{obs}}|\mathbf{X}, G),\end{aligned}$$

which provides a connection between the proposed generative framework and existing graph neural networks. The fourth term $-\eta \cdot \log q_\phi(\mathbf{Y}_{\mathrm{obs}}|\mathbf{X}, G)$ provides supervised information from labeled data for the approximate posterior GCN or GAT, while the other three terms can be viewed as additional regularizations: the learned GCN or GAT is encouraged to support the generative model of the graph, and not to go far away from $p_{\boldsymbol{\theta}}(\mathbf{Y}|\mathbf{X})$.

**Negative edge sampling.** For both LSM and SBM based models, the probability $p_{\boldsymbol{\theta}}(\mathbf{G}|\mathbf{X}, \mathbf{Y})$ factorizes to the production of probabilities of all possible edges. Calculating the log-likelihood of it requires enumeration of all possible $(i, j)$ pairs where $i, j \in \{1, 2, \cdots, n\}$, which results in a $\mathcal{O}(n^2)$ computational cost at each epoch. In practice, instead of going through all $(i, j)$ pairs, we only calculate the probabilities of the edges observed in the graph and a set of "negative edges" randomly sampled from the $(i, j)$ pairs where edges do not exist. This practical trick, named negative sampling, is commonly used in the training of word embeddings [15] and graph embeddings [20].

## 4 Experiments

In this section, we evaluate the proposed variants of $\mathrm{G}^3\mathrm{NN}$ on several benchmark datasets for graph-based semi-supervised learning. We test the models under both the standard benchmark setting [23] as well as two data-scarce settings.

### 4.1 Standard Benchmark Setting

We first consider a standard benchmark setting in recent graph-based semi-supervised learning literature [23, 10, 21].

**Datasets.** We use three standard semi-supervised learning benchmark datasets for graph neural networks, Citeseer, Cora, and Pubmed [19, 23]. The graph $G$ of each dataset is a citation network with documents as nodes and citations as edges. The feature vector of each node is a bag-of-words representation of the document and the class label represents the research area this document belongs to. We adopt these datasets from the PyTorch-Geometric library [4] in our experiments[5]. For each

Table 1: Summary of benchmark datasets.

| Dataset | # Classes | # Nodes | # Edges | Avg. 2-Neighborhood Size |
|---------|-----------|---------|---------|--------------------------|
| Cora | 7 | 2,708 | 5,278 | 35.8 |
| Pubmed | 3 | 19,717 | 44,324 | 59.1 |
| Citeseer | 6 | 3,327 | 4,552 | 14.1 |

Table 2: Classification accuracy under the standard benchmark setting. The upper block lists the discriminative baselines. The lower block lists the proposed variants of G$^3$NN. The **bold** marker denotes the best performance on each dataset. The underline marker denotes that the generative model outperforms its discriminative counterpart, e.g., LSM-GCN outperforms GCN; and the asterisk (*) marker denotes the difference is statistically significant by a t-test at significance level 0.05. The ($\pm$) error bar denotes the standard deviation of the test performance of 10 independent trials.

| | Cora | Pubmed | Citeseer |
|---|------|--------|----------|
| MLP | $0.583 \pm 0.009$ | $0.734 \pm 0.002$ | $0.569 \pm 0.008$ |
| GCN | $0.815 \pm 0.002$ | $\mathbf{0.794} \pm 0.004$ | $0.718 \pm 0.003$ |
| GAT | $0.825 \pm 0.005$ | $0.785 \pm 0.004$ | $0.715 \pm 0.007$ |
| LSM_GCN | $\underline{0.825} \pm 0.002*$ | $0.779 \pm 0.004$ | $\underline{0.744} \pm 0.003*$ |
| LSM_GAT | $\underline{\mathbf{0.829}} \pm 0.003$ | $0.776 \pm 0.007$ | $\underline{0.731} \pm 0.005*$ |
| SBM_GCN | $\underline{0.822} \pm 0.002*$ | $0.784 \pm 0.006$ | $\underline{\mathbf{0.745}} \pm 0.004*$ |
| SBM_GAT | $\underline{0.829} \pm 0.003$ | $0.774 \pm 0.004$ | $\underline{0.740} \pm 0.003*$ |

dataset, we summarize number of classes, number of nodes, number of edges, and average number of nodes within the 2-hop neighborhood of each node in Table 1. In this standard benchmark setting, we closely follow the dataset setup in Yang et al. [23] and Kipf and Welling [10].

**Models for comparison.** For the proposed framework, we implement four variants ($2 \times 2$) of G$^3$NN by combining the two generative model instantiations with the two approximate posterior model instantiations: LSM-GCN, SBM-GCN, LSM-GAT, SBM-GAT.

For baselines, we compare against two state-of-the-art models for the graph-based semi-supervised learning, GCN [10] and GAT [21]. We also include a multi-layer perceptron (MLP), which is a fully connected neural network without using any graph information, as a reference.

We use the original architectures of GCN and GAT models in both the baselines and the proposed methods. We grid search the number of hidden units from $(16, 32, 64)$ and the learning rate from $(0.001, 0.005, 0.01)$. GAT uses a multi-head attention mechanism. In our experiments, we fix the number of heads as 8 and try to set the total number of hidden units as $(16, 32, 64)$ and to set the number of hidden units of a single head as $(16, 32, 64)$. In the proposed methods, we set the generative model for $p_{\boldsymbol{\theta}}(\mathbf{Y}|\mathbf{X})$ as a two-layer MLP having the same number of hidden units as the corresponding GCN or GAT in the posterior model. For the MLP baseline, we also set the number of layers as 2 and grid search the number of hidden units and learning rate like other models. For the proposed generative models, we grid search the coefficient of the supervised loss $\eta$ from $(0.5, 1, 10)$. The number of negative edges is set to be the number of the observed edges in the graph. For LSM models, the dimensions of the feature transformation matrix $\boldsymbol{U}$ is fixed to $8 \times d$, where $d$ is the feature size. For SBM models, we use two settings of $(p_0, p_1)$: $(0.9, 0.1)$ and $(0.5, 0.6)$. We use Adam optimizer to train all the models and apply early stopping with the cross-entropy loss on the validation set. We adopt the implementations of GCN and GAT from the PyTorch-Geometric [4] library in all our experiments.

**Results.** The performance of the baselines and proposed models under the standard benchmark setting is summarized in Table 2. We report the mean and the standard deviation of the test accuracy of 10 independent trials for each model. The results show that on all datasets except for Pubmed, the proposed methods achieve the best test accuracy on the standard benchmark setting. Notably, every instantiation model of the proposed generative framework outperforms their corresponding discriminative baseline (GCN or GAT) in most cases. We also note that GCN performs better than GAT and the proposed models on Pubmed. We conjecture that, when the number of classes is

Table 3: Classification accuracy under the missing-edge setting. The **bold** marker, the underline marker, the asterisk (*) marker, and the ($\pm$) error bar share the same definitions in Table 2.

|  | Cora | Pubmed | Citeseer |
|---|---|---|---|
| MLP | $0.583 \pm 0.009$ | $0.734 \pm 0.002$ | $0.569 \pm 0.008$ |
| GCN | $0.665 \pm 0.007$ | $0.746 \pm 0.004$ | $0.652 \pm 0.005$ |
| GAT | $0.682 \pm 0.004$ | $0.744 \pm 0.006$ | $0.642 \pm 0.004$ |
| LSM_GCN | $\underline{0.711} \pm 0.005*$ | $\mathbf{\underline{0.766}} \pm 0.006*$ | $\underline{0.704} \pm 0.002*$ |
| LSM_GAT | $\underline{0.710} \pm 0.007*$ | $\underline{0.766} \pm 0.004*$ | $\underline{0.691} \pm 0.005*$ |
| SBM_GCN | $\mathbf{\underline{0.718}} \pm 0.004*$ | $\underline{0.762} \pm 0.005*$ | $\mathbf{\underline{0.716}} \pm 0.004*$ |
| SBM_GAT | $\underline{0.716} \pm 0.007*$ | $\underline{0.761} \pm 0.005*$ | $\underline{0.709} \pm 0.008*$ |

small and the graph is relatively dense, GCN may be already quite capable of propagating feature information from neighbors (see the average size of 2-hop neighborhoods in Table 1). When there are more classes or the graph is relatively sparse (e.g., Cora, Citeseer, and the missing-edge setting of Pubmed in Section 4.2.1), the advantage of our proposed method is more evident.

## 4.2 Data-Scarce Settings

Generative models usually have better sample efficiency than discriminative models. Therefore, we expect the proposed generative framework to show bigger advantage when data are scarce. Next, we evaluate the models on the citation datasets under two such settings: missing-edge setting and reduced-label setting.

### 4.2.1 Missing-Edge Setting

In the standard benchmark setting, we assume that all samples are connected to the graph identically and the training, validation, and test set are split randomly in the datasets. In practice, however, the samples we are interested in the test period may not be well connected to the graph. For example, we may not have connection for new users in a social network other than their profile information. In this cold-start situation, one may expect to make predictions purely based on the profile information. However, we believe that the relational information stored in the graph of the training data can still help us learn a better and more generalizable model even if some of the predictions are made only based on the node features. And we expect the proposed generative models to work better than the discriminative baselines in this case because they can better distill the relationship among the data.

To mimic such situations, we create a missing-edge setting, where we remove all the edges of the test nodes from the graph. Note that this setting is different from the inductive learning setting in previous works [5, 21] where the edges for the test data are absent during the training stage but present during the test stage. In the missing-edge setting, the edges for the test data are absent during both stages. We follow the same experimental setup as in the standard benchmark setting except for the modification of the graph.

**Results.** The performance under the missing-edge setting is shown in Table 3. Not surprisingly, as we lose part of the graph information, the performances of all models except for MLP (which does not use the graph at all) drop compared to the standard benchmark setting in Table 2. However, the proposed generative models perform better than their corresponding discriminative baselines by a large margin. Remarkably, even without knowing any edges of out-of-sample nodes, the accuracy of SBM-GCN on Citeseer can achieve the state-of-the-art level of GCN under the standard benchmark setting.

### 4.2.2 Reduced-Label Setting

Another common data-scarce situation is the lack of labeled data. Therefore, we create a reduced-label setting, where we drop half of the training labels for each class compared to the standard benchmark setting. All other experiment and model setups are the same as the standard benchmark setting.

**Results.** The performance under the reduced-label setting is shown in Table 4. As can be seen from the results, the proposed generative models achieve the best test accuracy on Cora and Citeseer again.

Table 4: Classification accuracy under the reduced-label setting. The **bold** marker, the underline marker, the asterisk (*) marker, and the ($\pm$) error bar share the same definitions in Table 2.

|          | Cora                  | Pubmed                | Citeseer              |
|----------|-----------------------|-----------------------|-----------------------|
| MLP      | $0.498 \pm 0.004$     | $0.674 \pm 0.005$     | $0.493 \pm 0.010$     |
| GCN      | $0.750 \pm 0.003$     | $\textbf{0.724} \pm 0.005$ | $0.666 \pm 0.003$ |
| GAT      | $0.771 \pm 0.004$     | $0.711 \pm 0.006$     | $0.675 \pm 0.005$     |
| LSM_GCN  | $\underline{0.777} \pm 0.002$* | $0.709 \pm 0.003$ | $\underline{0.691} \pm 0.005$* |
| LSM_GAT  | $\underline{0.792} \pm 0.004$* | $0.699 \pm 0.003$ | $\underline{0.691} \pm 0.004$* |
| SBM_GCN  | $\underline{0.780} \pm 0.002$* | $0.710 \pm 0.004$ | $\textbf{0.703} \pm 0.006$* |
| SBM_GAT  | $\textbf{0.796} \pm 0.008$* | $0.699 \pm 0.003$ | $\underline{0.698} \pm 0.003$* |

And the gap of the performances between the proposed generative models and the corresponding discriminative models are larger on Cora and Citeseer.

# 5 Conclusion

In this paper, we have presented a flexible generative framework for graph-based semi-supervised learning. By applying scalable variational inference, this framework is able to take the advantages of both recently developed graph neural networks and the wisdom of random graph models from classical network science literature, which leads to the $G^3NN$ model. We further implement 4 variants of $G^3NN$, instantiations of the proposed framework where we build generative graph models with graph neural networks as the approximate posterior models. Through thorough experiments, We demonstrated that these instantiation models outperform the state-of-the-art graph-based semi-supervised learning methods on most benchmark datasets under the standard benchmark setting. We also showed that the proposed generative framework has great potential in data-scarce situations.

For future work, we expect more complex instantiations of generative models to be developed using this framework and to optimize the graph-based semi-supervised learning.

**Acknowledgments**

This work was in part supported by the National Science Foundation under grant numbers 1633370 and 1620319.

## Footnotes

[5]The datasets loaded by the PyTorch-Geometric data loader have slightly less edges than those reported in Yang et al. [23], which is believed due to the existence of duplicate edges in the original datasets.

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
