[Reviews · NeurIPS 2019]

Reviewer 1



This work employs techniques developed in network science literature, such as latent space model (LSM) and stochastic block model (SBM), to propose a generative model for features X, outputs Y, and graph G, and it uses graph neural networks to approximate the posterior of missing outputs given X, observed Y, and G. This work is a wise combination of recent methods to effectively address the problem of graph-based semi-supervised learning. However, I have some concerns, which are summarized as follows: - Although the paper proposed a new interesting generative method for graph-based semi-supervised learning, it is not super novel, as it employs the other existing methods as the blocks of their method, like LSM, SBM, GCN, GAT. - It seems the generative model is only generative for G given X and Y and by factorizing the other part as p(Y,X) = p(Y|X) p(X), for p(Y|X), it is modeled via a multi-layer perceptron, which is a discriminative model. That is why the authors discard X in all the analyses, like any other discriminative model, and say that everything is conditioned on X. I think this makes the proposed model not fully generative. It is only generative for G but not for X and Y. I was wondering what would be the performance if you would assume p(G,Y,X) = p(G|X,Y) p(X|Y) p(Y), which makes the features X generative conditioned on the outputs? - It is not clarified how the parameters are learned from the ELBO. For example, in SBM, are p_0 and p_1 the only learnable parameters? If yes, how the constraints are taken into account? - Regarding the approximate posterior model, in part 3.3.2, the authors have used graph neural networks to approximate the posterior of missing Y given X, observed Y, and G. However, as mentioned in the paper, graph neural networks get only X and G as input but not any Y. It seems this is not a reasonable approximation as it is not consistent with the graph generation step, LSM and SBM, which use the label information to generate the graphs. What is the reasoning of using graph neural networks? Could you revise them to handle the labels too, which will make the approximation more realistic? - Having mediocre performance on Pubmed data might cast a doubt on dependence of the performance of the proposed method on the input data. How could you explain the poor performance on that dataset? Is there other datasets to test on to prove the efficacy of the proposed method?

Reviewer 2



Originality: The paper appears to be original. Quality: Good empirical results. It is reassuring that multiple instantiations of the model perform well and very valuable to show these results for different variations. Clarity: This paper is very clearly written and has excellent organisation. Significance: The authors provided a "simple" (yet powerful) implementation of their model, that is competitive with baseline models and may be a strong foundation for future work.

Reviewer 3



Originality – The proposed generative framework seems reasonable and new for graph-based semi-supervised learning. Quality – The proposed model still relies on GCN/GAT as inference models, and the experimental results only show marginal improvements over GCN/GAT on normal classification tasks. I find the experiments less convincing, and my main concerns are as follows. - 1. Experimental results should be compared against other generative models mentioned in section 2.3, especially Bayesian GCN [22], while the authors only make comparisons to the vanilla GCN/GAT. - 2. I am not sure the missing-edge setting (lines 248-253) is reasonable. The authors say they consider an “extreme” setting by removing all the edges of the test nodes from the graph. Since test nodes are the majority (> 95% in the experiments), after removing the edges connected to them, nearly no edges are left in the training set. How is it possible to train the graph model in this situation? Is it fair for GCN/GAT? Please correct me if I am wrong. - 3. The authors use a validation set with many labels to choose network hyperparameters and the balancing parameter \eta in the overall loss function and for early stopping. This contradicts with the label-scarce setting and is impractical. In fact, GCN/GAT can work reasonably well without using the validation set, albeit with a little performance drop. Bayesian GCN [22] also did not use the validation set. The experimental results would be more convincing if the validation set is not used for searching \eta and early stopping. Clarity – This paper is easy to follow and largely well written, but I think there is still room for improvement in writing and presentation. The notations should be kept consistent throughout the paper, e.g., the notation G of the graph is sometimes in bold and sometimes not. Significance – The proposed model still relies on GCN/GAT as inference models, and I don’t see a clear advantage of the proposed model over GCN/GAT except in the missing-edge setting, but this setting, as described in the paper, does not look reasonable or realistic to me. --------------------------------- After Rebuttal ------------------------------------------------------ I would like to thank the authors for their response. I have read their response and gone through the paper again carefully. I still have the following concerns. -1. Thanks for the explanation, now I understand the setup of the missing-edge experiment. But I still do not understand why the proposed method (GenGNN) would outperform GCN/GAT in this setting. In the training objective function (below line 177) of GenGNN, an additional ELBO loss is added to the loss of GCN/GAT to regularize their parameters. According to authors’ explanation in lines 181-182, this additional loss enforces the learned GCN/GAT to support the graph (G) generation while conforming to the label (Y) generation (by an MLP) without the graph. There seems to be some contradiction/tradeoff here, and I find it hard to comprehend. Why would the regularization help to train a better GCN/GAT for the missing-edge setting? The authors should provide better explanation than simply saying it would improve generalization as in line 244. Perhaps an ablation study would help to understand the contribution of each regularization term in the objective function? Also, the setting of the missing-edge experiment seems too “extreme”. To mimic the scenario of new users in a network, shouldn’t it be more reasonable to randomly set aside a small portion of nodes (as new users) instead of 1000 nodes (about 1/3 of the nodes in Cora/CiteSeer) for testing? For the experiments on normal graphs, although the proposed method could improve GCN/GAT on Cora and CiteSeer, it also consistently decreases their performance on PubMed for both the standard and reduced-label settings (similar to Bayesian GCN). Are the results on PubMed also statistically significant? The reason given by the authors in the paper and in their feedback says it is because the graph of PubMed is denser than that of Cora and CiteSeer, which sounds reasonable but vague. Like the missing-edge experiment, it would be more interesting to provide some insights from the training objective function while taking into account graph properties. -2. Thanks for the comparison to Bayesian GCN. In my opinion, it is OK to follow the hyperparameters of GCN/GAT, but the proposed method should be able to perform well (train reasonable model parameters) without using the validation set to ensure its practical use for semi-supervised learning. For GCN/GAT, I know they used the validation set in the original papers, but they can do well without using the validation set for model selection given 20 labels per class on the citation networks (verified in the experiments of Bayesian GCN and other papers). With fewer labels such as 10 or 5 per class, there will be a significant performance gap between with and w/o the validation set. Bayesian GCN did not use the validation set for model selection, which makes their results more convincing. It would be informative to show how the proposed method would perform without using the validation set for model selection/early stopping, especially for the reduced-label setting and the missing-edge setting. Other comments: There is a vast literature on semi-supervised learning, and graph neural networks is a fast-developing field. I find the listed references are a bit insufficient. It would be informative to include more recent works on GNN for discussion or comparison. My final assessment:I think the proposed generative model is valid and may have potentials, but for a NeurIPS paper, I would expect a clearer interpretation of model behavior or a more complete empirical evaluation. It would be better if the authors could clearly explain the benefits of GenGNN, i.e., in what scenarios it could improve GCN/GAT and more importantly WHY. So far, the message conveyed in this paper is not so clear, and the experimental results are not very convincing. I think this work still needs much revision to make a real impact/solid contribution to the field. Therefore, I tend to keep my previous rating.

[Author Response · NeurIPS 2019]

— For **Reviewer #1**. Thank you for the comments! We address your specific concerns in detail below. —

*Response to Q1*: We want to highlight that our main contribution is the novel flexible generative framework which takes
advantage of both network models and GNNs. Our technical contributions lie in the derivation of the variational method
when bridging the network models with GNNs. We intend to use simple existing building blocks to avoid unnecessary
confounding factors for proof-of-concept of the general framework, as mentioned in lines 142 to 144.

*Response to Q2*: First, the model we used, $P(G, Y, X) = P(G|X, Y)P(Y|X)P(X)$ is as much a "fully generative
model" as $P(G|X, Y)P(X|Y)P(Y)$. Note that specifying $P(X)$ only results in an extra additive term to the loss at
line 177 (together as $L(\theta, \phi) - log(P(X))$), which is irrelevant to the model we will use for prediction. Therefore,
removing this term will lead to no loss of generality. We agree $P(G|X, Y)P(X|Y)P(Y)$ could be another design
choice for the generative model. We did try both and found the one in our paper works better at this task.

*Response to Q3*: For LSM models, the ELBO is fully differentiable w.r.t. all model parameters, therefore the parameters
can be learned through SGD. For the simple instantiation of SBM used in this paper, $p_0$ and $p_1$ are the only learnable
parameters of $P(e_{ij}|y_i, y_j)$. In principle, they can also be learned through SGD, but given just two parameters, we
found grid-search with limited combinations (line 220) works better in practice. In future work, we will attempt to
further improve the estimation of the parameters in the SBM, taking advantage of the vast literature in this domain. As
proof-of-concept, the simple learning procedure is sufficient to obtain desirable improvements on the benchmarks.

*Response to Q4*: We acknowledge that the GNN-based approximate posterior models do not accommodate $Y$. We made
this choice mainly as a trade-off for computational efficiency. Even with this approximation, the GNNs can leverage the
generative model and improve the inference performance. Interestingly, we recently became aware of a concurrent
study (Graph Markov Neural Networks, ICML 2019) after our submission which used the same approximation (see
their Eq.3). We plan to design more inclusive model structures that can efficiently handle the labels in future work.

*Response to Q5*: We did further analysis to investigate the reasons. First, note GAT is no better than GCN on Pubmed.
Second, the number of classes is smaller on Pubmed (3) than on Cora (7) and Citeseer (6). The average number of 2-hop
neighbors is much larger on Pubmed (57.1) than on Cora (35.0) and Citeseer (13.5). When the number of classes is
small and the graph is relatively dense, GCN is already quite capable of propagating feature information from neighbors,
which makes it difficult to further improve. When there are more classes or the graph is relatively sparse (e.g., Cora,
Citeseer, and the missing-edge setting of Pubmed), the advantage of our proposed method is more evident.

— For **Reviewer #2**. Thank you for the encouraging comments! —

— For **Reviewer #3**. Thank you for the comments! We address your specific concerns in detail below. —

*Response to Q1*: We agree that Bayesian GCN (B-GCN) should have been another proper baseline. However, the
code of B-GCN is not released and we were not able to reproduce the results in [22] despite our due diligence. We
instead test LSM_GCN closely following the experimental setup of [22] and compare with the results reported in [22].
Specifically, we use fixed hyper-parameters (HPs) for LSM_GCN, where those of the GCN part are the same as B-GCN.
We also fix the LSM part with hidden size 16 and $\eta = 1$. To assure fair comparison, we use the official (fixed) split
and report averaged results of 50 runs. As an evidence of fair comparison, below we report the GCN performances
from [22] and from our implementation, which are very similar. The results show that LSM_GCN outperforms both
B-GCN and GCN in most cases. Besides the empirical comparisons, we also want to highlight that when it comes to
more complex graphs, the proposed framework (modeling the relationship between the graph, node labels, and features)
can better utilize the information of the data than B-GCN (modeling the graph alone).

| Methods | Cora 20 | Cora 10 | Citeseer 20 | Citeseer 10 | Pubmed 20 | Pubmed 10 |
|---|---|---|---|---|---|---|
| GCN ([22]) / GCN (ours) | 81.6 \| 81.5 | 74.9 \| 74.8 | 70.8 \| 71.4 | 65.8 \| 66.8 | 78.9 \| 79.0 | 72.8 \| 71.7 |
| B-GCN([22]) / LSM_GCN | 81.2 \| 82.4 | 76.6 \| 78.6 | 72.2 \| 73.8 | 70.8 \| 68.7 | 76.6 \| 78.0 | 72.3 \| 70.6 |

*Response to Q2*: First, we clarify that the test data (Planetoid split) in each dataset has only 1,000 nodes, which account
for about 1/2 of the edges in Cora and Citeseer, and 1/9 in Pubmed. Second, as we mentioned in 4.2.1, this setting
is realistic and actually has important practical value: these types of predictions are usually applied to new users in
social media networks who are likely to have few or no connections. There are a large number of low degree nodes in
real world networks given the power-law degree distribution. Finally, this setting signifies another advantage of the
proposed method over B-GCN: B-GCN cannot transfer knowledge to isolated nodes as it models the graph alone.

*Response to Q3*: Combining the experiments in the response to Q1 and the experiments in our paper, we have evaluated
our model and baselines in two ways: with and without tuning the HPs using the validation set. In both ways, our
method outperformed the baselines in most cases. We also want to clarify that both the GCN and the GAT papers did
use the validation set (see their experiment section for details), and B-GCN borrowed the HPs from the GCN paper.

[Meta-Review · NeurIPS 2019]

This paper proposes a generative framework for graph-based semi-supervised learning for approximating the joint distribution of the graph structure, labels and the node features. Variational inference techniques are then used to approximate the Bayesian posterior. The paper is well written. There are some issues raised by reviewer 3 regarding a better positioning of GenGNN with respect to GCN/GAT; which are recommended to be taken into account for the final version of the paper.